# Obesity–An Update on the Basic Pathophysiology and Review of Recent Therapeutic Advances

**DOI:** 10.3390/biom11101426

**Published:** 2021-09-29

**Authors:** Erind Gjermeni, Anna S. Kirstein, Florentien Kolbig, Michael Kirchhof, Linnaeus Bundalian, Julius L. Katzmann, Ulrich Laufs, Matthias Blüher, Antje Garten, Diana Le Duc

**Affiliations:** 1Department of Electrophysiology, Heart Center Leipzig at University of Leipzig, 04289 Leipzig, Germany; erind_g@yahoo.com; 2Department of Cardiology, Median Centre for Rehabilitation Schmannewitz, 04774 Dahlen, Germany; michael.kirchhof@median-kliniken.de; 3Pediatric Research Center, University Hospital for Children and Adolescents, Leipzig University, 04103 Leipzig, Germany; anna.kirstein@medizin.uni-leipzig.de (A.S.K.); florentien.kolbig@medizin.uni-leipzig.de (F.K.); antje.garten@medizin.uni-leipzig.de (A.G.); 4Institute of Human Genetics, University Medical Center Leipzig, 04103 Leipzig, Germany; linnaeusbundalian@gmail.com; 5Klinik und Poliklinik für Kardiologie, University Clinic Leipzig, 04103 Leipzig, Germany; julius.katzmann@medizin.uni-leipzig.de (J.L.K.); ulrich.laufs@medizin.uni-leipzig.de (U.L.); 6Helmholtz Institute for Metabolic, Obesity and Vascular Research (HI-MAG) of the Helmholtz Zentrum München at the University of Leipzig and University Hospital Leipzig, 04103 Leipzig, Germany; matthias.blueher@medizin.uni-leipzig.de; 7Department of Evolutionary Genetics, Max Planck Institute for Evolutionary Anthropology, 04103 Leipzig, Germany

**Keywords:** obesity, diabetes mellitus, anti-obesity drugs, obesity metabolism, energy balance

## Abstract

Obesity represents a major public health problem with a prevalence increasing at an alarming rate worldwide. Continuous intensive efforts to elucidate the complex pathophysiology and improve clinical management have led to a better understanding of biomolecules like gut hormones, antagonists of orexigenic signals, stimulants of fat utilization, and/or inhibitors of fat absorption. In this article, we will review the pathophysiology and pharmacotherapy of obesity including intersection points to the new generation of antidiabetic drugs. We provide insight into the effectiveness of currently approved anti-obesity drugs and other therapeutic avenues that can be explored.

## 1. Introduction

The leading causes for death and disability in the western world are chronic conditions like diabetes, cardiovascular disease, and cancer [1,2,3,4,5], which are closely linked to obesity. Body mass index (BMI) (weight in kg/height in m^2^), the most used formula to define overweight (BMI 25 to 29.9 kg/m^2^) and obesity (BMI ≥ 30 kg/m^2^), is simple to use in health screenings and epidemiological surveys. The relation between BMI and clinical outcomes has been extensively analyzed and there is near universal acceptance of ranges of BMI consistent with good health.

Obesity dramatically increases the risk for type 2 diabetes and both conditions represent major public health issues worldwide according to the latest reports from World Health Organization (WHO report 2020 and 2021). Also, obesity is considered to be the second most common, and may soon become, the most common preventable cause of cancer, overtaking cigarette smoking [6].

The incidence of obesity has tripled in the last few decades, such that more than two thirds (70.2%) of the United States adult population is overweight or obese and almost half of adults (48.5%) live with prediabetes or diabetes [5,6,7]. The pandemic proportions of obesity are still rapidly rising, challenging our concept of normality [8,9].

Currently recommended therapies with evidence-based support are lifestyle intervention, pharmacotherapy, and bariatric surgery [2]. While clinical practitioners occupy a crucial role in the front line of obesity- and its related comorbidities management, they receive minimal training in obesity management [2,10] leaving them ill-equipped to address the environmental and socioeconomic drivers of the obesity pandemic.

Significant scientific efforts consisting in elucidation of the complex physiopathology, as well as the clinical management of obesity and weight-related comorbidities such as type 2 diabetes (T2D), have materialized in the last few years in effective and safe treatment options [2,3,5,11,12,13,14,15,16,17,18]. The unprecedented amount of data to analyze in order to identify mechanisms and new targets can, though, easily become overwhelming. The pathophysiologic mechanisms of these metabolic conditions, lying at the crossroad of different highly specialized medical fields such as genetics, cellular and molecular biology, endocrinology etc. can only now be unveiled by taking advantage of ‘omics’ technologies, which may finally lead to a precision medicine approach.

Weight loss is the most important factor to reduce comorbidities and T2D in obesity; however, except for bariatric surgery, which can only be used in a minority of patients and can lead to significant complications, other therapy options have not been sufficiently effective [4,19]. The role of pharmacotherapy in the management of obesity has not been exploited in clinical practice, mainly because of the moderate weight loss effects and side effects of previous weight loss medications [20]. More recently developed anti-obesity therapies promise to overcome previous concerns about low effect size and safety concerns [6,21,22,23,24,25,26,27,28].

In the present review we deliver an update focused on the pathophysiology and pharmacotherapy of obesity, including currently approved drugs and other potential therapeutic targets.

## 2. Obesity Pathomechanisms

In a simplistic view weight gain, and ultimately obesity, results from a long-term positive energy balance, yet the pathogenesis of obesity has been proven to be more complex than this [29]. There is an intricate interplay between genetic, environmental, and psychosocial factors which mediate food intake and energy expenditure [30]. While the environment and socioeconomic conditions influence the behavior and cannot be molecularly tackled, the identification of genes and molecules that determine the susceptibility to obesity uncovers pathophysiological mechanisms, which can be molecularly addressed.

Studies on twins and families have estimated the rate of BMI heritability to be fairly high reaching 40–70% [31,32]. Large-scale genome-wide association studies have identified more than 300 loci bearing common variants in the general population, which show a significant correlation with obesity traits [30]. However, the effects of these loci on the obesity risk is fairly small and can explain less than 5% in the BMI variation [33,34]. Whether the missing heritability can be explained by epigenetic processes or interactions between genetic and environmental factors remains to be investigated by newly developed branches of research.

Research into monogenic obesity, where rare variants exert very large effects, has underlined the importance of biomolecules in the pathogenesis of obesity. Mapping mutations which cause extreme obesity in mice proved to be a successful strategy in the identification of monogenic disorders. Prominent outcomes of this approach include the identification of genes involved in body weight homeostasis, which act in the central nervous system: e.g., leptin (*Lep*) and its receptor (*Lepr*), the melanocortin 4 receptor (*Mc4r*) and pro-opiomelanocortin (*Pomc*) [35]. Pathogenic variants in the human orthologous genes lead to monogenic obesity [36]. Thus, these were the first milestones in understanding the mechanisms that govern hunger and satiety. Energy balance is controlled by complex interactions between the central nervous system, adipose tissue, and a plethora of other organs including the gut, liver, and pancreas (Figure 1).

Already in the 1940s it was apparent that hypothalamus plays an important role for energy metabolism regulation [37]. The hypothalamus integrates signals reflecting long-term energy stores and short-term nutritional input, which result in control of food intake, physical activity, and basal energy expenditure [38]. Short-term eating behavior can additionally be controlled by the hindbrain where the nucleus of the *tractus solitarius* (NTS) receives input from vagus nerve afferent, which is stimulated by secretin (SCT) and cholecystokinin (CCK) [39] (Figure 1).

Appetite-stimulating neurons in the arcuate nucleus (green) contain neuropeptide Y (NPY), which stimulates Y receptors (Y1 and Y5), and agouti-related peptide (AgRP), an antagonist of MC3/4 receptor activity. Brain peptides that stimulate appetite are NPY, AgRP, and endocannabinoids. Ghrelin is released by the stomach and has an orexigenic effect.

### 2.1. Molecules Acting on Long-Term Energy Balance

The main players for regulating energy balance as result of long-term energy stores are leptin and insulin.

#### 2.1.1. Leptin

Leptin is a 167 amino acid hormone secreted by white adipose tissue (i.e., adipokine), which circulates at concentrations proportional to body fat mass. It promotes satiety and energy expenditure by stimulating proopiomelanocortin (POMC) and inhibiting neuropeptide Y (NPY)/Agouti-related peptide (AgRP) neurons in the hypothalamus. A deficiency of leptin signaling as a result of mutations of the leptin gene or its cognate receptor causes hyperphagia and severe obesity in both humans and animals [40], which clearly demonstrates that normal body-weight requires intact leptin regulation.

Although genetic defects affecting leptin signaling cause obesity, such individuals are fairly rare [41]. Usually, obese individuals display increased leptin levels proportional to their body fat content. This raised the possibility that obesity may be associated with a form of “leptin resistance”. This would imply the stimulating threshold of leptin is increased and hence higher levels are needed to curb food intake and increase energy expenditure. Yet, in diet-induced obesity leptin cellular signaling appears to be intact [42], suggesting that the higher leptin levels are not a result of resistance, but rather some individuals need higher levels in order to engage the neuronal circuits.

Alternatively, it has been proposed that leptin could be more relevant for preventing loss of body fat, rather than reducing fat accumulation. In this setup only a decrease in leptin below the threshold of appetite inhibition becomes relevant and increased leptin circulating levels cannot further reduce energy intake [43]. More research is needed to fully understand leptin’s role in common obesity [44].

#### 2.1.2. Insulin

Insulin is secreted by pancreatic β-cells. Its levels are also positively correlated with body weight and adipose mass, and they provide a negative feedback signal to the central nervous system. Thus, like leptin, high levels of insulin result in reduced food intake. Obesity is characterized by insulin resistance and hyperglycemia, commonly accepted to be caused by increased levels of free fatty acids, which ultimately results in hyperinsulinemia [45]. Several studies have suggested that increased insulin secretion contributes to obesity pathogenesis by stimulating the adipocyte uptake of fatty acids and glucose and the caloric storage in form of fat, while concomitantly inhibiting lipolysis [43].

Dietary carbohydrates, refined sugars in particular, have been suggested to increase insulin secretion [43]. It would thus be tempting to assume that by replacing carbohydrates with fat, the effects of hyperinsulinism can be counteracted, which confers obesity protection. However, several observations have challenged this hypothesis. An analysis of weight-loss diets [46] showed that although low carbohydrate, higher fat diets led to slightly greater weight loss than did low-fat diets (~1 kg), the difference was fairly small and one diet could not be recommended over the other. Low-carbohydrate diets may increase energy expenditure and thus contribute to the maintenance of a reduced body weight [47], but differences in protein content of the alternative diet could confound the results. It thus becomes very clear that body fat accumulation cannot be tackled from only one direction but should be addressed rather in a systemic fashion.

#### 2.1.3. Proopiomelanocortin (POMC)

Adjacent to NPY/AgRP neurons, in the arcuate hypothalamic nucleus some neurons express POMC and release α-, β- and γ-MSH (melanocyte-stimulating hormone). α-MSH is a potent anorexic neuropeptide that reduces food intake by stimulating melanocortin 4 receptors (MC4R) expressed on “downstream” target neurons from the paraventricular hypothalamic nucleus (Figure 1). In conditions of leptin deficiency POMC neurons are inhibited, while NPY neurons are stimulated resulting in hyperphagia by releasing the break in both directions [48].

Since POMC targets the melanocortin system (Figure 1), it is clear why mutations that impair this system (e.g., MC4R) determine hyperphagic obesity in both humans and animal models [48]. POMC neurons are viewed as the counterpoint of AgRP neurons. However, unlike AgRP neurons which affect appetite very fast, POMC neurons are extremely slow in affecting hunger (many hours). To close this loop recently a subset of oxytocin receptor-expressing excitatory neurons that powerfully and rapidly inhibit feeding and are modulated by α-MSH have been described [49]. Future studies are still needed to integrate all these circuits in the big picture of energy balance.

### 2.2. Molecules Leading to Short-Term Positive energy Balance–Orexigenic Stimuli

#### 2.2.1. Neuropeptide Y

A major role in energy homeostasis is attributed to NPY/AgRP neurons. These are a subset of neurons found in the arcuate nucleus that can synthesize both NPY and AgRP [50].

NPY is a neuropeptide composed of 36 amino acids that is involved in numerous physiological processes both in the central and peripheral nervous systems. It is one of the most powerful controllers of feeding and energy homeostasis regulation and is highly expressed in the central nervous system. In the brain it is produced in the arcuate nucleus, and is the most potent short-term stimulus for appetite [40]. NPY is also produced by neurons of the sympathetic nervous system and induces vasoconstriction and fat tissue expansion. Negative energy balance leads to an elevation of hypothalamic NPY levels triggering an increase in food intake (Figure 1) and a simultaneous decrease in energy expenditure mostly by inhibiting sympathetic output. However, NPY signaling also influences a variety of other physiological processes that are linked to altering mood and anxiety thereby limiting its potential use as a clinically feasible target for appetite inhibition and energy expenditure intervention [50,51].

#### 2.2.2. Agouti-Related Peptide

AgRP is also an appetite stimulating neuropeptide, which acts on the same neurons as NPY (Figure 1). AgRP neurons are activated in conditions of negative energy balance (e.g., fasting), characterized by decreased plasma concentrations of leptin and insulin, that tonically inhibit these neurons [52]. Selective AgRP neuron activation elicits hyperphagia to counteract the state of metabolic need [53]. Activation of these neurons shifts the energy balance towards intake either by stimulating rewarding mechanisms associated with food or by reducing the discomfort associated with not eating. Whether the mechanism favors reward promotion or discomfort alleviation is still controversial. Hence, the study by Chen et al. suggests that as long as food is available, AgRP neuron activation is highly rewarding [54], while Betley et al. propose that AgRP neuron activation is aversive when food is not available [55]. Thus, the psychological effect of AgRP neuron activation may be related to food availability.

#### 2.2.3. Ghrelin

Another hormone that reaches highest plasma levels during fasting and immediately before meals, similar to AgRP, is ghrelin [56]. Ghrelin is synthesized by cells located throughout the gastrointestinal tract, at highest density in the fundus of the stomach [44,57] (Figure 1). In the arcuate nucleus of the hypothalamus, ghrelin activates the same neurons as NPY and AgRP [58] (Figure 1) and stimulates appetite. Additionally, ghrelin also stimulates growth hormone release [59].

Several human genetic studies identified rare mutations and single nucleotide polymorphisms (SNPs) in the gene encoding ghrelin receptor, which might be associated with human obesity and short stature [60]. Moreover, two SNPs in the receptor (Ala204Glu and Phe279Leu) have been associated with obesity and very short height [61]. Yet, the role of ghrelin in obesity is still not clarified, since knockout mice of both the ghrelin gene [62] and of the ghrelin receptor [63] do not display a distinguishable phenotype. Also, obese individuals usually display low ghrelin plasma levels and show an increased secretion only after weight loss [64]. However, ghrelin administration in cancer patients with anorexia increased the energy intake, suggesting it may be a good option for anorexia treatment [65]. Thus, the effect of ghrelin stimulation may be dependent on the overall endocrine milieu, making it hard to establish a direct role in obesity.

#### 2.2.4. Endocannabinoids

One of the effects of cannabis (*Cannabis sativa*) consumption is increased appetite [66]. However, the discovery of the endocannabinoid system (ECS) [67], the receptors and its endogenous ligands, has substantiated the central role ECS plays in governing appetite, ingestive behavior, energy metabolism, and body weight [66]. There are two receptors CB1 and CB2, with CB1 being distributed throughout the brain, primarily in the hypothalamus and limbic system, which are involved in the regulation of food intake and its rewarding capacities [67]. CB2 is on the other hand mainly present in immune cells and it is believed to play a role in immunogenicity [68]. Research suggests still there may be additional receptors to the ones already known [66]. CB1 knockout mice do not develop diet-induced obesity or insulin resistance on high-fat diet, although they are only slightly hypophagic [69]. This suggested that CB1 is implicated in peripheral metabolic regulations. Indeed, peripheral treatment with the CB1 antagonist (Rimonabant) in rats activated lipid mobilization pathways in white adipose tissue and cellular glucose uptake and interestingly, also reduced food intake and body weight [70]. Understanding the exact mechanisms of the CB1 peripheral stimulation appears to be of clinical relevance and future research is warranted.

### 2.3. Molecules Leading to Short-Term Negative Energy Balance–Anorexigenic Stimuli

#### 2.3.1. Secretin

SCT is a 27-amino acid peptide produced by S-cells in duodenum as a response to acid. SCT is mostly known for bicarbonate secretion in the pancreas. Both the peripheral (via vagal activation) and central SCT can induce an anorectic effect without causing conditioned taste aversion [71]. Feeding induced increase in circulating secretin activates brown adipose tissue thermogenesis by stimulating lipolysis through binding to secretin receptors in brown adipocytes, which is sensed in the brain and promotes satiation. The anorectic effect of peripheral and central SCT was confirmed experimentally in mice. Intracerebroventricular and intraperitoneal injections of SCT reduced food intake in wild-type mice, but not in SCT-Receptor knockout mice [71,72]. Research on food intake modulations by SCT has been initiated for the past few years, but still requires further efforts to test its potential in therapeutic interventions.

#### 2.3.2. Cholecystokinin

CCK is the prototype of a satiety hormone produced by mucosal enteroendocrine cells of the duodenum and jejunum, neurons from the enteric nervous system, and the brain [73]. The secretion of CCK is stimulated by the presence of food in the gut lumen and leads to meal termination [74].

CCK activates vagal afferent neurons that convey the gastrointestinal signal to hindbrain areas, including the nucleus of the solitary tract [75] (Figure 1). Some of these hindbrain neurons project to the parabrachial nucleus, a central hub involved in appetite regulation. A variety of stimuli linked to food intake, like gastric distention, secretion of CCK or glucagon-like peptide 1 (GLP-1) activate calcium gene-related peptide expressing neurons in the parabrachial nucleus (CGRP^PBN^) which leads to the physiological meal termination [76]. Activation of these neurons has also been implicated in rapid-onset severe, life-threatening anorexia [77]. Because hypothalamic AgRP neurons inhibit CGRP^PBN^ neurons, AgRP stimulation appears to promote feeding, in part, by inhibiting CGRP^PBN^ neurons [76]. Yet, unlike what is observed in mice with defective melanocortin signaling, CGRP^PBN^ neuron inactivation does not result in diet-induced obesity [77]. Whereas POMC neurons play a physiological role to limit food intake over long time intervals, CCK-stimulated CGRP^PBN^ neurons appear to constitute an acute break to food consumption during individual meals [43].

#### 2.3.3. Incretin Hormones

Incretin hormones are peptides secreted by specialized entero-endocrine cells at different gut levels in response to food intake leading to stimulation of insulin secretion. The known incretin hormones are GIP (glucose-dependent insulinotropic polypeptide) and GLP-1 (glucagon-like peptide-1). They are responsible for a two- to three-fold higher insulin secretory response to oral as compared to intravenous glucose administration, a phenomenon called the incretin effect [78]. In addition to their insulinotropic activity, incretin hormones also affect glucagon release. GIP stimulates glucagon secretion [78] especially at lower glucose concentrations, while GLP-1 suppresses glucagon secretion [78,79], in particular at hyperglycemia. GIP can also lead to increased insulin-stimulated glucose transport, fatty acid synthesis, and incorporation into triglycerides [40].

GLP-1, in addition, has significant effects on multiple organ systems. Most relevant are a reduction in appetite and food intake, leading to weight loss in the long term.

Both GIP and GLP-1 are rapidly degraded by the enzyme dipeptidyl peptidase IV (DPP-IV), leading to a circulating half-life of only 2 min for GLP-1 [80]. Long-acting DPP-IV–resistant GLP-1 agonists reduce food intake [81], while the GLP-1 antagonists leads to increased food intake [82]. However, the mouse knockout of the GLP-1 receptor does not develop obesity [83]. Conversely, knockout of GIP receptor protects against diet induced obesity by increasing energy expenditure [84].

The two hormones also have multiple additional effects in adipose cells, bone, and the cardiovascular system. Recently, multiple clinical outcome trials have shown that GLP-1 receptor agonists reduce cardiovascular adverse events and prolong life in patients with type 2 diabetes [3,13,17,18,85,86]. The incretin system currently provides an important anti-obesity therapeutic target.

#### 2.3.4. Oxyntomodulin

Preproglucagon gene product yields two important satiety peptides, GLP-1 and oxyntomodulin (OXM). Like GLP-1, OXM is released from entero-endocrine cells in response to nutrients in the form of free fatty acids and carbohydrates. OXM activates both glucagon-like peptide-1 receptor (GLP1R) and the glucagon receptor (GCGR) resulting in reduced food intake [87,88] and increased energy expenditure [89]. Interestingly, although both GLP-1 and OXM activate GLP1R, GLP1 stimulates mainly areas in the brainstem [90], while OXM acts on the arcuate nucleus [91]. Moreover, studies on *Glp1r* knockout mice suggested that the ligand-specific activation of the receptor results in differentially potent inhibition of appetite and increase of energy expenditure [92]. However, the differential effect of OXM vs. GLP1 may be explained by the additional activation of the GCGR through OXM [89].

Nevertheless, preclinical studies demonstrated the anti-obesity effect of OXM since a chronic administration leads to superior weight loss and comparable glucose lowering to a GLP1R-selective peptide [93,94]. OXM acutely improves glucose metabolism in both rodents [95] and humans by significantly increasing insulin secretion and lowering glucose levels even in type 2 diabetes patients [96]. Thus, targeting OXM in therapy could prove beneficial not only for addressing obesity, but also its comorbidities.

#### 2.3.5. Polypeptide Fold (PP-Fold) Family

The PP-fold family consists of neuropeptide Y (NPY), peptide tyrosine tyrosine (PYY), and pancreatic polypeptide (PP), which are all 36 amino acids hormones. These hormones are important mediators along the gut-brain axis and act via four subtypes of G-protein-coupled receptors (Y1, Y2, Y4, Y5) [97]. NPY is, as detailed above, the most potent orexigenic stimulus, mainly expressed in the arcuate nucleus in hypothalamus.

PP and PYY, on the other hand, are gut-derived peptides released after food ingestion with an anorexigenic effect [98]. PYY is mainly produced by intestinal enteroendocrine L cells, while PP is synthesized by endocrine F cells of the pancreatic islets [99]. The release of both PP and PYY occurs proportionally to the caloric intake [100] and results in the inhibition of the hypothalamic orexigenic pathways [101]. PYY acts via Y2 receptors and inhibits the release of NPY in the arcuate nucleus [102] and PP activates vagal cholinergic pathways in the brainstem via Y4 receptors [103]. In obese patients intravenous infusion of physiological levels of PYY reduces the caloric intake [104]. Similarly, PP reduces appetite and food intake in healthy human volunteers [105]. However, chronic peripheral administration of PP in lean and obese mice reduces caloric intake, but central administration leads to increases caloric intake [106]. Such disparities, also seen with OXM and GLP-1 effects, may be caused when stimulating different receptors. E.g. for PP stimulation of Y4 receptors in the area postrema can reduce food intake, while activation of Y5 receptors expressed elsewhere in the brain leads to increase food intake [40].

Nonetheless, obese patients usually display reduced circulating PYY [104] and PP [107] as well as elevated NPY [108] levels which suggests a role of the PP-fold family in the pathophysiology of obesity.

#### 2.3.6. Amylin

Amylin is secreted by pancreatic β cells and co-released with insulin [109]. It is also expressed in the lateral hypothalamus where it acts synergistically with leptin to reduce energy intake [110]. In addition to leptin function, amylin increases energy expenditure and it influences hedonic aspects of eating that may lead to food type selection [111,112]. While amylin acts together with leptin in the arcuate nucleus, the hindbrain area postrema is the one critically involved in mediating the satiating effect [113]. Here, amylin increases cGMP [114] and phosphorylates ERK [115], signals that trigger the anorexic effect.

Recent studies suggest that amylin can influence the rewarding properties of food [116]. Direct activation of amylin receptors in rats reduces not only the intake of low palatable chow, but also the intake of a palatable sucrose solution [117]. Additionally, central administration of amylin reduces reward-driven behaviors like lever pressing for sucrose solution in rats, suggesting that both the consummatory phase and the appetitive phase of eating are affected [118]. Thus, these results have prompted the amylin-based pharmacotherapy as detailed below.

#### 2.3.7. Cocaine- and Amphetamine-Regulated Transcript

Cocaine- and amphetamine- regulated transcript (CART) is a biologically active peptide that has emerged as a potent regulator of food intake and energy balance in mammals. It is widely distributed in the brain of mammals and plays a role in neuroendocrine and autonomic regulation, controls physiological and behavioral functions, including feeding inhibition and anxiety stimulation [119,120]. Intracerebroventricular administration of CART potently suppresses food intake in fed and fasting animals and CART expression in the brain is reduced during fasting. In the vertebrate hierarchy, CART and NPY appear as a classic example of two signaling agents which have coevolved to exert antagonistic effects in energy homeostasis [119].

### 2.4. White Adipose Tissue Secreted Molecules

The adipose tissue is no longer considered to be only an inert storage tissue, but rather a metabolically dynamic organ capable of producing biologically active molecules involved in metabolic homeostasis. A large body of evidence shows that adipokines play a role in feeding modulation, glucose and lipid metabolism, or inflammatory and immune functions.

To this end, the adipokines that were first discovered were involved in appetite modulation— leptin or insulin sensitivity—adiponectin, resistin. Adiponectin was one of the earliest adipokines described [121]. This peptide is secreted by adipocytes in large quantities and, counterintuitively, circulating levels are lower in obese compared to lean subjects [122]. A reduced adiponectin secretion has been suggested to play a central role in obesity-related diseases, including insulin resistance/type 2 diabetes, and cardiovascular disease [123]. Based on animal models adiponectin was suggested to be an essential regulator for health- and lifespan [124]. The enhanced insulin sensitivity primarily mediated by adiponectin is a result of increased fatty acid oxidation and suppressed hepatic glucose production [125]. Although, the main function of adiponectin is peripheric, adiponectin and leptin have a synergistic action on the brain and both promote weight loss [126]. However, while leptin inhibits the appetite, adiponectin increases energy expenditure.

The counterpart of adiponectin is resistin, an adipose-derived hormone expressed in adipocytes in rodents and, in humans, mainly in peripheral blood mononuclear cells [127]. The molecule was named “resistin” to be suggestive for the induced insulin resistance and impaired glucose tolerance observed in mice that underwent treatment with a recombinant form [128]. The structure and secretion levels of resistin in humans differ from that of rodents, such that several functions might also vary between the species. Thus, the role of resistin in inducing insulin resistance is controversial in humans. Several studies found positive correlations between resistin and insulin resistance [129,130], while others failed to find any correlations between resistin and insulin resistance [131,132]. A recent meta-analysis suggests that type 2 diabetes and obesity do not necessarily need to be associated with resistin, but when resistin displays high circulating levels insulin resistance occurs [133]. Future research is necessary to decide on the utility of targeting resistin for obesity interventions.

More recently, other adipokines as chemerin, lipocalin-2 (LCN2), vaspin, and omentin-1 have been implicated in neuroendocrine-immune interactions and inflammatory reactions [134], while for many other newly discovered adipokines (reviewed here [134]) the underlying molecular mechanisms are still not entirely clear.

### 2.5. Brown Adipose Tissue (BAT) Secreted Molecules

While white adipose tissue is the main energy storage tissue in the body, brown adipose tissue has a distinct role in non-shivering thermogenesis. Expression of uncoupling protein-1 (UCP1) in brown adipocytes allows dissociating electron transport by the mitochondrial respiratory chain from ATP production, which results in heat generation. The high energy consumption of this process makes brown adipose tissue an interesting target for obesity therapy and there are many studies in animals [135,136,137] and humans [138,139,140] pointing to the benefits of increasing brown adipose tissue to increase energy expenditure, reduce body weight or reduce insulin resistance. So called “browning” of white adipose tissue takes place within white adipose tissue depots when adipocyte precursors are induced to differentiate towards the brown adipocyte phenotype by various secreted factors [141]. Among these, β3-adrenergic receptor agonists seemed an attractive approach to induce browning of white adipose tissue. Clinical studies in the early 1990ies failed, however, due to limited efficacy or side effects [142] and none was approved for obesity therapy so far.

More recent research points to the role of factors secreted by brown adipose tissue, so called brown adipokines or BATokines, in regulating systemic metabolism [143,144]. Among these, fibroblast growth factor 21 (FGF21) is secreted from BAT upon thermogenic activation [145,146] and induces the browning of white adipose tissue in mice [147]. Its cardioprotective effects [148] and its impact on activation of lipolysis [149] make FGF21 an interesting target for obesity intervention. C-X-C motif chemokine ligand-14 (CXCL14) is a BATokine that promotes browning via recruitment of M2 macrophages to white adipose tissue [150]. Several factors from the bone morphogenic protein (BMP) family are involved in regulation of adipocyte development. While BMP7 and BMP8B promote brown adipocyte differentiation, BMP4 induces the differentiation from brown into white adipocytes in some contexts [151,152,153]. Meteorin-like (Mtrnl) is a hormone secreted upon cold exposure or exercise and increases energy expenditure and thermogenesis in white adipose tissue by recruitment of alternatively activated macrophages [154]. The secretion of neuregulin-4 (NGR4) from white and brown adipose tissue is also induced upon cold exposure and low levels of NGR4 are associated with diabetes in rodents and humans [155,156]. Current knowledge on these and other BATokines and their therapeutic potential was recently reviewed [144,157].

### 2.6. Obesity, Mitochondrial Dysfunction and Insulin Resistance–An Interplay

Mitochondria are central to adipocyte function since they are the organelles where key metabolic pathways, such as beta oxidation and ATP production, take place (Figure 2). Accordingly, mitochondrial dysfunction in adipocytes has a significant impact not only on adipocytes themselves, but also on whole body energy metabolism [158]. Excess energy uptake as one reason for the development of obesity affects mitochondrial function in multiple ways. Peroxisome proliferator-activated receptor coactivator-1 α (PGC-1α) is a master regulator of mitochondrial biogenesis by interacting with mitochondria-related transcription factors and a major inducer of mitochondrial oxidative metabolism [159]. Under conditions of energy excess, PGC-1α is acetylated and silenced [160,161]. A subsequent reduction in oxidative phosphorylation results in an accumulation of metabolic intermediates, i.e., diacylglycerols (DG) and ceramides (CER) [162] that may contribute to the development of insulin resistance via an inhibition of insulin receptor signaling mediated through an interaction with Proteinkinase C (PKC) in case of DG or with AKT in case of CER. Another major contributor to mitochondrial dysfunction is elevated level of free fatty acids that lead to increased generation of Reactive Oxygen Species (ROS) during mitochondrial beta oxidation (Figure 2). Having the potential to damage cellular structures, ROS can induce mitochondrial mitophagy and apoptosis which results in a decrease of mitochondrial number and exacerbation of lipid accumulation with consequent adipocyte hypertrophy [163]. Hypertrophic adipocytes secrete proinflammatory cytokines, such as monocyte chemoattractant protein 1 (MCP-1), whereupon proinflammatory macrophages are attracted to the adipose tissue depot, contributing further to an impaired adipose tissue function and chronic low-grade inflammation [164,165], which is associated with impaired insulin sensitivity and the development of type 2 diabetes [166] (Figure 2).

## 3. Pharmacotherapy Options

Therapeutic interventions based on lifestyle and diet changes have shown only modest results [167]. Thus, there is growing interest in drug therapy that can support and promote weight loss. Generally, pharmacotherapy either enhances satiety and inhibits hunger or increases catabolism. As detailed above increasing the understanding of the underlying pathophysiology of obesity holds promise for developing new more potent medications to curb the obesity epidemic.

Currently there are six major anti-obesity medications approved by the United States Food and Drug Administration (FDA) (Table 1, Figure 3). Of these as of August 2021 the European Medicines Agency (EMA) has approved only four substances, since the combination therapy phentermine/topiramate was rejected in 2013 and semaglutide is under evaluation since January 2021 and up to the date of the current review.

Except for orlistat, liraglutide, and the recently FDA-approved semaglutide the other approved drugs influence only central nervous system pathways that either reduce appetite or enhance satiety (Figure 3).

### 3.1. Approved Anti-Obesity Drugs for Long-Term Weight Management

#### 3.1.1. Orlistat

Available since 1999 in many countries, orlistat is one of the anti-obesity drugs approved by both the FDA and EMA. It is a selective inhibitor of pancreatic lipase, it acts only in the periphery, and decreases fat absorption by 30% [12,168].

Efficacy: Orlistat is indicated in conjunction with a reduced-calorie diet for patients with a BMI ≥ 30 kg/m^2^ or ≥ 28 kg/m^2^ with comorbidities like hypertension, diabetes, hyperlipidemia [38]. A systematic review observed a mean weight loss of 3.1 kg/year associated with orlistat therapy [169]. Orlistat was shown to reduce the incidence of type 2 diabetes [170] and of LDL and total cholesterol regardless of the weight-loss impact [171].

Safety: Gastrointestinal side effects, reduced absorption of fat-soluble vitamins and steatorrhea are very frequent [38]. Orlistat is also over-the-counter at a lower dose of 60 mg compared with the prescription-compulsory 120 mg formulation, suggesting no major side effects are of concern [172].

Clinical insight: Orlistat 120 mg is administered three times daily. Although not severe the adverse effects which include flatulence or fecal urgency and only a modest average weight loss make orlistat less popular than the appetite-suppressants described in the following sections [172].

#### 3.1.2. Naltrexone/Bupropion

Naltrexone is an opiate antagonist, which blocks opioid receptor-mediated POMC auto-inhibition, whilst bupropion selectively inhibits reuptake of dopamine and noradrenaline. The combination therapy is approved since 2012 by both the FDA and EMA. The combination promotes satiety via enhancement of hypothalamic POMC-mediated release of melanocyte-stimulating hormone (MSH) resulting in reduced food intake and increased energy expenditure [173].

Efficacy: Four major 56-week phase III randomized, placebo-controlled trials have evaluated the efficacy [174,175,176,177]. A meta-analysis reported an annual weight loss of 4.8% total body weight (mean 4.4 kg) [169].

Safety: Common adverse events include nausea, constipation, headaches, vomiting, dizziness, and dry mouth [172]. The drug is not recommended for patients with a history of seizures, drug addiction, and bulimia and/or anorexia nervosa. Patients should not combine the medication with other opiates [172].

Clinical insight: The current naltrexone SR/bupropion SR combination is available in 8 mg naltrexone SR and 90 mg bupropion dosing, which is upwards titrated over a 4-week period for a total dosage of two tablets twice daily: one tablet once daily in the morning during week 1, one tablet twice daily during week 2, two tablets in the morning and one tablet in the evening during week 3, and finally two tablets twice daily during week 4 [172]. During the initial stage, if significant adverse effects occur, the dose should not be further escalated until better tolerability is established. If within 12 weeks less than 5% of initial body weight is lost the drug should be discontinued. Naltrexone SR/bupropion SR appears to have good effects in patients with food addiction [177] and binge-eating disorder concomitant with alcohol abuse [178].

#### 3.1.3. Phentermine/Topiramate

Phentermine/topiramate is an extended-release combination which was FDA-approved in 2012. EMA refused its marketing authorization in Europe in 2013 raising concerns with respect to cardiovascular safety and adverse psychiatric effects [179]. Phentermine is a sympathomimetic that stimulates noradrenaline and suppresses appetite. Topiramate is an antiepileptic also used to treat migraine. The exact anorexigenic mechanism of topiramate is not known; its effect of appetite suppression is postulated to result from modulation of various neurotransmitters, like inhibition of voltage-dependent sodium channels, glutamate receptors, and carbonic anhydrase and the potentiation of γ-aminobutyrate activity [180]. The combination of the two drugs has greater weight-loss- and lower side-effects than monotherapy with each one [172].

Efficacy: Two large randomized, double-blind, placebo-controlled trials over 52 weeks [181,182] and one 2-year extension trial with an additional 52 weeks of treatment have been conducted [183]. A recent meta-analysis noted a mean weight loss of 9.8 kg per year in the randomised-controlled trials [169]. Patients also benefitted from improved lipid profile, glycemic control, and waist circumference [181,182,183].

Safety: While EMA has not approved the usage, FDA expressed concerns in respect to teratogenicity, cardiovascular side effects (triggered by phentermine), as well as cognitive, psychiatric, and metabolic acidosis (this three being triggered by topiramate) [172]. Abrupt withdrawal of topiramate increases the risk of seizures, thus gradual downward titration of the combination drug, over 3–5 days is recommended [181,182,183]. Common side effects include insomnia, dizziness, and paresthesia [38].

Clinical insight: This combination drug shows one of the highest weight-loss effects, but the price of $232/kg weight loss can be a limiting factor [184]. The individual active substances can be prescribed separately in available monotherapy dosages (phentermine 8 mg, 15 mg, 30 mg and 37.5 mg; topiramate 15 mg, 25 mg, 50 mg, 100 mg and 200 mg) [172]. Topiramate monotherapy is not approved for the treatment of obesity, but it has shown benefits for treating binge-eating disorder [185] and against weight regain following bariatric surgery [186].

#### 3.1.4. Liraglutide

Liraglutide is a GLP1 receptor agonist of the incretin hormone which affects glucose homeostasis, food intake, and satiety. It is approved since 2012 at 1.8 mg for T2D and at a higher dose of 3.0 mg for obesity. Due to the incretin properties, the drug was originally marketed for the treatment of type 2 diabetes [38]. Liraglutide and semaglutide are the only approved injectable anti-obesity drug, while all others are oral drugs [172].

Efficacy: The efficacy of 3.0 mg liraglutide in combination with a reduced-calorie diet and increased physical activity was assessed in four 56-week randomized, placebo-controlled trials [187,188,189,190]. A meta-analysis noted an additional annual weight loss of 5.3–5.9 kg compared to placebo [169].

Safety: Most common side effects are transient and mild to moderate intensity gastrointestinal symptoms like: nausea, diarrhea, constipation, vomiting, dyspepsia, and abdominal pain [187,188,189,190]. Liraglutide is contraindicated in patients with a family or personal history of medullary thyroid carcinoma or patients with multiple endocrine neoplasia type 2 (MEN2) syndrome [172]. The reason for the contraindication is that rats and mice exposed to liraglutide developed malignant thyroid C-cell carcinomas; however the implication for humans has not been established yet [191]. There appears to be an increased risk for pancreatitis, although some studies did not find this to be statistically significant [192]. A recent meta-analyses addressing this issue followed patients up to 3 years and concluded that GLP-1 receptor agonists including liraglutide and semaglutide can be used with no safety concerns related to malignant neoplasia [193].

Clinical insight: The initial dose is 0.6 mg subcutaneously once daily for the first week followed by 0.6 mg increments every week, to a maximum of 3.0 mg. In case of persisting adverse effects the dose is not increased until better tolerability is achieved [172].

#### 3.1.5. Setmelanotide

Setmelanotide is an MC4R agonist, which acts in the paraventricular nucleus of the hypothalamus and in the lateral hypothalamic area to suppress the appetite [38]. It has been recently approved by FDA (November 2020) [194] and EMA (July 2021) [195] for the treatment of monogenic forms of obesity.

Setmelanotide is approved starting age six in patients with obesity in the presence of genetic variants in *POMC*, *PCSK1*, or *LEPR* genes classified as pathogenic, likely pathogenic, or of uncertain significance [194] according to the American College for Medical Genetics Guidelines [196].

Efficacy: The treatment of two patients with POMC deficiency resulted in massive weight loss of 51.0 kg and 20.5 kg over 42 and 12 weeks, respectively [196]. A clinical trial including obese participants with heterozygous MC4R deficiency and setmelanotide lead to a placebo-adjusted weight loss of 2.6 kg, which is far less than in individuals with POMC defects [197]. There is a recently completed clinical trial (8 March 2021), which evaluated the effect of setmelanotide in patients with Bardet-Biedl and Alström Syndromes suffering from obesity [198]. Results from this study are not published at the time of the present manuscript writing (August 2021). A long-term trial including 150 participants with genetic defects upstream of the MC4 Receptor (Figure 1) in the leptin-melanocortin pathway is expected to finish in March 2023 [199].

Safety: Common side effects include injection site reactions, skin hyperpigmentation, headache, and gastrointestinal symptoms like nausea, diarrhea, and abdominal pain [194].

Clinical insight: Setmelanotide is administered subcutaneously with a starting dose of 2 mg/day for two weeks. If the initial dose is not tolerated, the dose should be reduced to half till the desired tolerability is achieved. If the initial dose is well tolerated it can be titrated to 3 mg/day. If weight loss is not ≥5% of baseline body weight after 12–16 weeks of treatment, the administration should be discontinued.

#### 3.1.6. Semaglutide

Semaglutide, the most recently approved anti-obesity drug, is another GLP1 receptor agonist with similar mechanism of action as liraglutide. It was initially designed as a potent long-acting drug that could be administered up to 1 mg subcutaneously once weekly, and was approved for treatment of type 2 diabetes in 2017 [200]. In June 2021 FDA approved semaglutide at a higher dose (2.4 mg) once weekly [25] for chronic weight management in obese or overweight adults with at least one weight-related condition. It is also currently under review by EMA.

Efficacy: The efficacy and safety of semaglutide 2.4 mg/week versus placebo was evaluated in four 68-week trials [21,22,23,24]. In three of the trials including patients without diabetes mellitus the treatment group attained an almost incredible 15%–18% weight loss (corresponding to a placebo-subtracted 10.6–15.8 kg) over 68 weeks. In one trial including only patients with type 2 diabetes and a BMI of at least 27 kg/m^2^, the mean weight loss was a placebo-subtracted of 6.2%. These findings are seen as a major breakthrough for obesity treatment and the drug is considered a game-changer [26].

Safety: Semaglutide is generally well-tolerated. The most common side effects were nausea, diarrhea, vomiting, constipation, abdominal pain, headache, fatigue, dyspepsia, dizziness, abdominal distension, eructation, hypoglycemia for diabetic patients, flatulence, gastroenteritis, and gastroesophageal reflux disease. Serious adverse events were reported in 9.1%and 2.9% of the participants in the semaglutide and placebo groups respectively, including hepatobiliary disorders and infections [24]. Like liraglutide, there is a warning for potential risk of thyroid C-cell tumors. It is contraindicated for patients with a personal or family history of medullary thyroid carcinoma or MEN2 [25].

Clinical insights: Semaglutide is administrated once weekly, initiated at 0.25 mg, with dose escalation every 4 weeks to 0.5 mg, 1 mg, 1.7 mg until the target dose of 2.4 mg/week is reached. If participants cannot tolerate the 2.4-mg dose, they can receive 1.7 mg instead. They should be encouraged to make at least 1 attempt to re-escalate to the 2.4-mg dose.

### 3.2. Other Drugs in Potential Use for Anti-Obesity Treatment

#### 3.2.1. Lorcaserin (Withdrawn from Market in February 2020)

Lorcaserin is a 5-hydroxytryptamine receptor 2C (5-HT_2c_) agonist that acts on anorexigenic POMC neurons in the hypothalamus to suppress the appetite. It was approved for use by the FDA in 2012 as support for diet and lifestyle changes in adults with a BMI ≥30 kg/m^2^ or those with a BMI ≥ 27 kg/m^2^ and at least one weight-related comorbid condition such as diabetes, hypertension, hyperlipidemia or sleep apnoea [201]. Three major randomized, double-blind, placebo-controlled trials of 52-week or 104-week duration evaluated the efficacy of lorcaserin [202,203,204]. A systematic review noted a mean annual weight loss of 3.1 kg [169] additional to improved metabolic parameters like blood pressure, total and LDL cholesterol [205].

However, despite good weight-loss results EMA did not approve the use of lorcaserin and FDA requested withdrawal from market in February 2020 because of possible increased risk of colorectal, pancreatic, and lung cancer.

#### 3.2.2. Sodium-Glucose Co-Transporter-2 Inhibitors (Not Indicated for Obesity Alone)

Inhibitors of Sodium-Glucose Co-Transporter-2 (SGLT2) were initially introduced as treatment option for type 2 diabetes mellitus. In healthy individuals the glucose filtered at the renal glomerulus is almost completely reabsorbed up to a threshold of 10 mmol/L (180 mg/dL) of plasma glucose concentration [206,207]. The SGLT2 is responsible for the glucose reabsorption of >90% of the glucose filtered at the glomerulus [208]. SGLT2 inhibitors lead to ca. 50% of filtered glucose to be excreted, thus improving glycemic control and reducing body mass typically by 2 kg [209]. In large clinical trials an unexpected and consistent lowering of the heart failure risk was observed. These findings led to the initiation of dedicated heart failure trials [15,206,210,211]. Both dapagliflozin and empagliflozin were shown to reduce heart failure (HF) risk independently of the presence of type 2 diabetes in patients with reduced ejection fraction (HFrEF) [3]. Large cardiovascular outcome trials have also shown significant prognosis improvement in patients with HFrEF when treated with canaglifozin or sotaglifozin [15,206,212,213,214]. While several treatment options to improve prognosis of patients with HFrEF are available, for patients with heart failure with preserved ejection fraction (HFpEF) there are currently none available. Very recently however, a landmark study on the treatment of HFpEF has been published. In the EMPEROR-Preserved trial, treatment with empaglifozin resulted in a 21% reduction of the composite endpoint of cardiovascular death or hospitalization for heart failure [215].

Additionally, in a DAPA-CKD trial, treatment with dapaglifozin resulted in a 39% reduction of the composite endpoint of glomerular filtration rate, a decline of 50% in end-stage kidney disease, or death from renal or cardiovascular causes. Based on these results, in April 2021 FDA approved dapagliflozin also for treatment of patients with chronic kidney disease, irrespective of whether they have diabetes or heart failure. Similar results have been reported in patients with diabetes treated with canagliflozin.

The complete spectrum of mechanisms that lead to clinical benefits are currently extensively investigated because they are unlikely to be related to the improved glycemic control alone. The reduction in heart failure risk and renoprotective effect are likely the result of a combined effect of different mechanisms. Owing to the increased glucose excretion, SGLT2 inhibitor use is associated with improved glycemic control, body mass reduction, and a reduction in systemic blood pressure (typically of 4 mmHg systolic). Also, natriuresis and a reduction in plasma volume are likely to be protective against the development of HF and might explain at least part of the rapid-onset reduction in the risk of hospitalization for HF. The increased distal renal sodium delivery, thereby increasing glomerular afferent arteriolar tone and reducing hyperfiltration can contribute to the renoprotective effect. Epicardial fat is intensely metabolically active and secretes profibrotic and pro-inflammatory cytokines, which can adversely affect the myocardium and coronary arterial tree. Treatment with SGLT2 inhibitors is associated with reduced pericardial adipose tissue deposition. Also, an in vitro study showed a reduced secretion of pro-inflammatory chemokines by the epicardial fat. Moreover, SGLT2 inhibition reduces glucose oxidation, increases fat oxidation and increases plasma concentrations of ketone bodies. A shift towards ketone body production has been proposed as a potential mechanism for the rapid cardiovascular benefit. In addition, the increase in lipid oxidation might reduce the levels of toxic intracellular lipid metabolites leading to a reduction in cardiac steatosis. [216,217] In a genetic analysis, SNPs in the SGLT2-encoding gene were associated with reduced risk of heart failure, while the main mediators of this association were changes in HbA1c, HDL cholesterol, and uric acid [213]. Further research is required to provide a deeper understanding of the mechanisms of action.

Treatment with SGLT2 inhibitors is not indicated for the treatment of obesity. However, there is an indication and approval for common co-morbidities of obesity such as type 2 diabetes and heart- and renal failure. Since SGLT2 inhibitors beneficially influence many factors that contribute to the development of complications of obesity it appears justified to assume that patients with obesity and an indication for SGLT2 inhibitors will especially benefit from treatment with a member of this drug class.

### 3.3. Cardiovascular Comorbidities Outcomes for Anti-Obesity Medication

While cardiovascular outcomes trials are ongoing with anti-obesity agents, none of the drugs at the dose and indication to treat obesity has shown to reduce major cardiovascular adverse events (MACE) [6]. However, following weight loss, many anti-obesity drugs improve the cardiovascular risk factors. Also, in patients with T2DM SGLT2 inhibitors and GLP1-RA (in the dose indicated for diabetes treatment) can reduce MACE, and in some cases, reduce overall mortality [3,5,6,28]. Ongoing cardiovascular outcome studies are evaluating subcutaneous semaglutide 2.4 mg per week in patients with obesity (SELECT-Trial).

### 3.4. Further Potential Pharmacotherapeutic Targets

#### 3.4.1. Polyagonists of the Incretin System

A drug that affects multiple receptors may have a greater weight loss effect compared with single mechanism molecules. This has led to the development of several incretin-related therapies such as GLP-1/glucagon receptor dual agonists, GLP-1/GIP dual agonists, and GLP-1/GIP/glucagon receptor triple agonists [218,219]. As detailed above GLP1 controls glycemic regulation and induces satiety. Glucagon receptor stimulation via oxyntomodulin enhances energy expenditure by increasing oxygen consumption, lipid catabolism, and thermogenesis [220]. GIP increases pancreatic insulin release, and in low glucose state it also stimulates glucagon secretion [38]. GIP analogues can thus potentiate the diabetogenic glucagon effect but given the opposing effect of GLP1 on glucagon secretion, the combined therapy showed better outcomes. While these drugs are not yet approved for use preclinical and recent clinical trials show promising results [218,219,221].

#### 3.4.2. Amylin Mimetics

Amylin induces satiety and inhibits glucagon secretion. The amylin analogue pramlintide was licensed by the FDA in 2005 for patients with insulin-treated diabetes [38]. In diabetic patients the amylin analogue can induce weight loss by curbing the appetite [222]. However, results with different analogues appeared to be quite variable, which lead to discontinuation of most mono- and combination therapies [38].

#### 3.4.3. Leptin Analogues

Metreleptin is a recombinant analog of human leptin with an additional methionine compared to the leptin amino acid sequence. While based on the energy balance physiology, detailed above, there is a strong rationale for leptin analogues in obesity treatment, metreleptin failed to achieve clinically meaningful weight loss with a mean of just 1.5 kg lost over 24 weeks [223]. To enhance the effect of leptin analogues, a combination therapy with amylin mimetics such as pramlintide was tested with good results showing 11.5 kg weight loss over 20 weeks compared with 7.4 kg and 7.9 kg for metreleptin or pramlintide monotherapy, respectively [44]. However, following commercial reassessment the further development of the combination therapy was discontinued in 2011 [38]. Still, in 2014 FDA approved the use of metreleptin in patients with leptin deficiency or lipodystrophy as subcutaneous injection in a once or twice daily administration [38].

#### 3.4.4. Ghrelin Vaccine and Antagonists of Ghrelin and NPY

Ghrelin is the only known hormone secreted by the digestive system with an orexigenic effect. As detailed above, ghrelin stimulates NPY- and inhibits POMC-neurons leading to increased appetite. An interesting therapy approach was vaccination against ghrelin, which slowed weight gain in rats by decreasing feeding efficiency [224]. Also ghrelin antagonists promoted weight loss and improved insulin tolerance in rats [225], but both strategies showed no success in human studies and clinical studies are currently not undertaken [38,226]. Similarly, NPY inhibition did not achieve clinically meaningful results to justify obesity treatment [227]. It appears thus that inhibiting the appetite stimulants reaches less effective results compared to the stimulation of anorexic stimuli mediators.

#### 3.4.5. Cannabinoid Type-1 Receptor Antagonists

Rimonabant, a CB1 antagonist showed excellent weight loss outcomes with an additional mean weight loss of 4.7 kg in clinical trials [228]. Similar to the animal models [70], rimonabant use improved metabolic markers including glycemic and lipid control [229]. Yet, in 2009 three years after EMA approval, the drug had to be withdrawn because of the increased risk of severe mood disorders and suicide [230]. Based on the peripheral effects of CB1 antagonists in rats [70] it appears of clinical relevance to design less hydrophobic CB1 antagonists to reduce the blood-brain barrier penetration and therefore the centrally mediated psychiatric side effects [38].

#### 3.4.6. Antioxidants

Obesity has been linked to lower concentrations of different antioxidants, such as Vitamin C, Vitamin E, and glutathione [231,232,233], as well as increased oxidative stress induced by macrophages which release proinflammatory cytokines (Figure 2). This disbalance in the redox state not only alters lipid and carbohydrate metabolism, by promoting insulin resistance [232,233], but also controls appetite and weight by influencing the satiety and hunger mechanisms in the hypothalamus [234]. This suggests that antioxidants can be beneficial in obesity treatment [231,234].

Indeed, lipoic acid, a cofactor of pyruvate dehydrogenase complex, which is involved in energy metabolism, has been used as a supplement in obesity therapy [231], with a good outcome on weight, BMI, and inflammation markers. Procyanidins are polyphenols with antioxidant function, found in vast amounts in cinnamon [231]. In a study, cinnamon extract showed promising results on fasting blood glucose levels, oxidative stress, and fat mass [231]. Catechins, another type of polyphenols, present in green teas, lead to fat oxidation and increased thermogenesis [231]. Initial promising results in this field may impose antioxidant molecules as necessary supplements for obesity treatment, whether by pharmacologic, surgical or just lifestyle interventions.

## 4. Conclusions

The Obesity Medicine Association defines obesity as a chronic, progressive, relapsing, and treatable multi-factorial, neurobehavioral disease, wherein an increase in body fat promotes adipose tissue dysfunction and abnormal fat mass physical forces, resulting in adverse metabolic, biomechanical, and psychosocial health consequences [6]. The complicated definition testifies for the complexity of the problem. The more simple and pragmatic definition of overweight and obesity, as a body mass index (BMI) over 25 and 30 respectively is nearly universally accepted. Obesity represents a rapidly escalating public health issue, taking over many parts of the world [85,235] that contrary to conventional belief is not limited to industrialized countries [235], with the majority of affected children living now in developing countries [85]. Considering the worldwide number of individuals affected, there are good arguments to see obesity as the biggest pandemic of all time. Thus, it is high time to develop preventive and therapeutic strategies. An improved understanding of the pathophysiology, including fat redistribution and accumulation is a prerequisite for developing more potent therapies. Insights gained from animal [236] and cellular [237,238,239] models for obesity may eventually help identify additional relevant pathways and molecules, which can ultimately lead to better treatments. In this article we have highlighted known players in the physiology of energy balance and the corresponding potential or already approved therapies. The rapidly evolving diabetes pharmacotherapy will surely contribute to the introduction of new drugs to tackle this pandemic. Considering the history of other metabolic diseases (i.e., hypertension, dyslipidemia and diabetes mellitus) if the ongoing trials provide evidence that anti-obesity agents can improve MACE, their acceptance and use in clinical practice can increase dramatically.

## Figures and Tables

**Figure 1 biomolecules-11-01426-f001:**
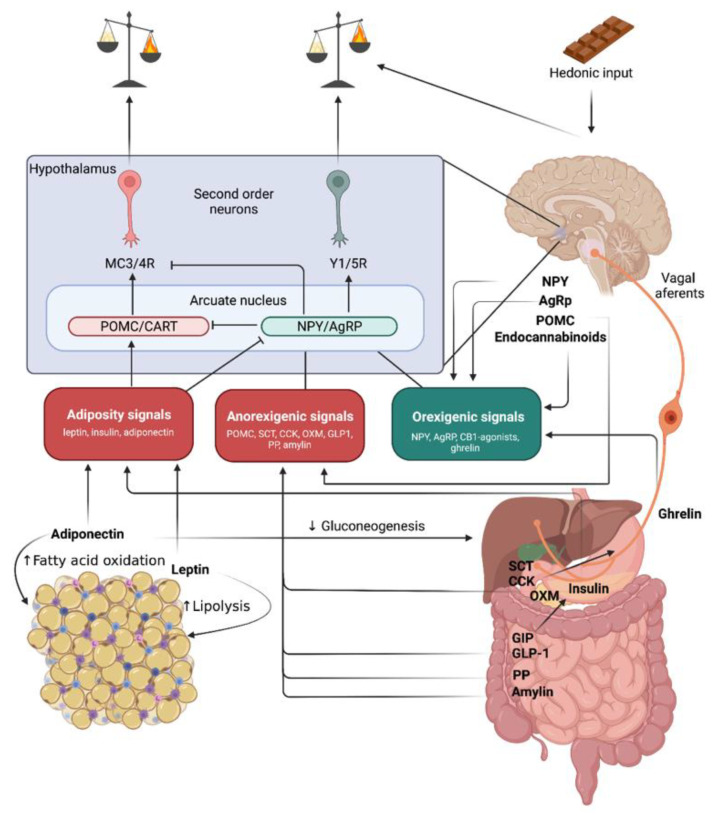
Energy balance signals integration. In the blue quadrant there is a simplified representation of hypothalamic energy balance regulation mechanisms: primary neurons in the arcuate nucleus include appetite-inhibiting neurons (red)–cocaine-and amphetamine-stimulated transcript peptide (CART) and proopiomelanocortin (POMC), which release peptides that stimulate the melanocortin receptors (MC3 and MC4). MC3/4R stimulation increases energy expenditure and decreases appetite. This circuit is stimulated by adiposity and anorexigenic signals. Peripheral signals related to long-term energy stores are produced by adipose tissue (leptin, adiponectin) and the pancreas (insulin). Gut hormones with incretin-, hunger-, and satiety-stimulating effects: glucagon-like peptide-1 (GLP-1), glucose-dependent insulinotropic polypeptide (GIP), and potentially oxyntomodulin (OXM) improve the response of the endocrine pancreas to absorbed nutrients; GLP-1 and OXM also centrally reduce food intake; secretin (SCT) and cholecystokinin (CCK) released from the gut inhibit appetite by way of the vagus nerves, which stimulate hindbrain structures.

**Figure 2 biomolecules-11-01426-f002:**
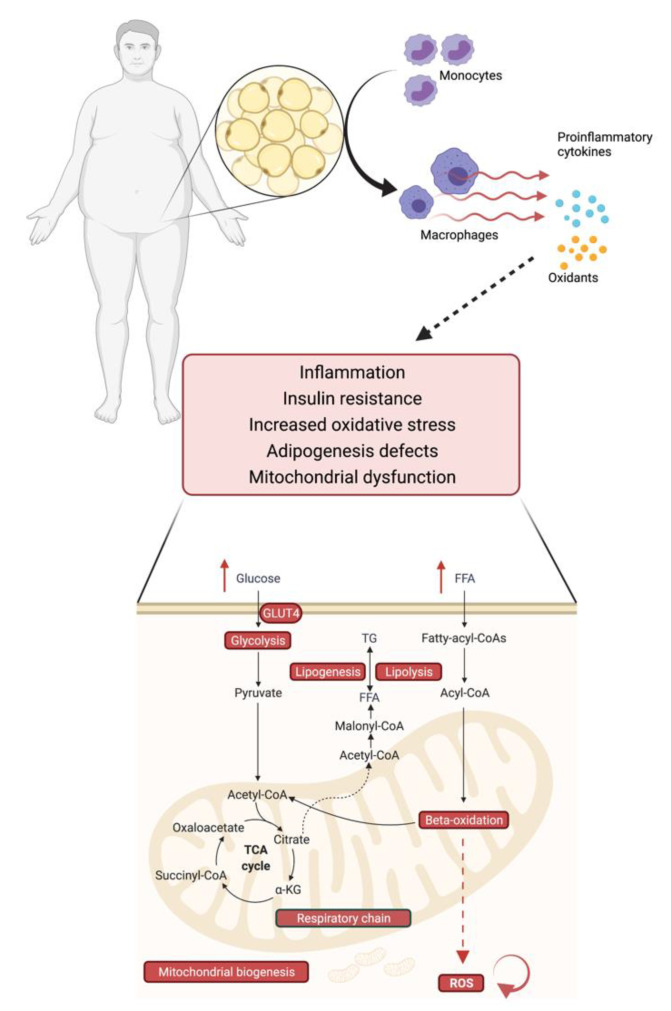
Obesity is linked to mitochondrial dysfunction and insulin resistance. In obesity, proinflammatory macrophages are attracted to the adipose tissue depot and release proinflammatory cytokines, which trigger, amongst other reactions, inflammation, increased oxidative stress, and mitochondrial dysfunction. Dietary nutrient excess leads to an overload of mitochondria with free fatty acids (FFA) and glucose, which impairs multiple mitochondrial functions (in red). The reduced beta oxidation leads to an increase in reactive oxygen species (ROS), which reinforces the mitochondrial damage and contributes to insulin resistance and adipose tissue dysfunction.

**Figure 3 biomolecules-11-01426-f003:**
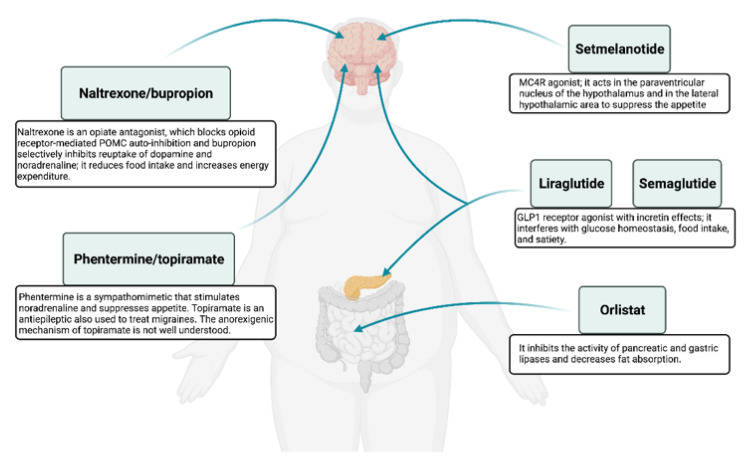
Summary of the mechanism of action for FDA/EMA approved anti-obesity drugs.

**Table 1 biomolecules-11-01426-t001:** Overview of the FDA/EMA approved pharmacotherapy options.

Drug	Mean Weight Loss at ≥1 year, Placebo-Subtracted	Side Effects	Precaution
Phentermine/topiramate	9.8 kg	insomnia, dizziness, paresthesia, depression, anxiety, memory problems	abrupt withdrawal of topiramate increases the risk of seizures
Naltrexone/bupropion	4.4 kg	nausea, constipation, headaches, vomiting, dizziness, dry mouth	not recommended for patients with seizures, drug addiction, bulimia, anorexia nervosa or in combination with opiates
Liraglutide	5.3–5.9 kg	nausea, diarrhoea, constipation, vomiting, dyspepsia, abdominal pain	contraindicated in patients with a family or personal history of medullary thyroid carcinoma or with MEN2 syndrome (rats and mice developed thyroid C-cell carcinomas; unclear implication for humans)
Semaglutide	6.6–15.8 kg	nausea, diarrhea, vomiting, constipation, abdominal pain, headache, fatigue, dyspepsia, dizziness, hypoglycemia for diabetic patients, flatulence, gastroenteritis	potential risk of thyroid C-cell tumors. It is contraindicated for patients with a personal or family history of medullary thyroid carcinoma or MEN2.
Orlistat	3.1 kg	vitamin deficiency, steatorrhea, fecal urgency, fecal incontinence	daily multivitamin intake is recommended because of malabsorption of fat-soluble vitamins
Setmelanotide	2.6 kg in MC4R deficiency, 51 and 20.5 kg in 2 patients with POMC deficiency	injection site reactions, skin hyperpigmentation, headache, nausea, diarrhea, abdominal pain	approved for monogenic forms of obesity

## Data Availability

Not applicable.

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
