# Peer review of "Obesity–An Update on the Basic Pathophysiology and Review of Recent Therapeutic Advances"

_biomolecules, 2021, doi:10.3390/biom11101426_

Round 1

Reviewer 1 Report

In the current manuscript entitled “Obesity – an update on the basic pathophysiology and review of recent therapeutic advances”, Gjermeni and collaborators review the main pathological mechanisms involving the development of obesity, including a clear summary of the main endocrine factors and their action on the CNS, participating in the regulation of energy balance. The authors also include an analysis of the state-of-the-art on the currently approved pharmacological strategies to treat obesity.

The review is well-structured and provides very complete information, and under my opinion, very few criticisms can be done, although I would like to suggest some recommendations to clarify some aspects of the manuscript, especially involving the section “Adipose tissue secreted molecules”:

  • In the introductory paragraph, I would suggest that authors mention that the adipokines described in that section are only some of the several adipokines currently described (i.e. adipokines not included are omentin-1, SFRP5, cardiotrophin, FABP4, lipocalin-2, apelin…) and the reason of this selection.
  • I intuit the reason why authors highlight the importance of leptin in regulating energy balance and they do not include it in this section, but authors should clearly indicate that leptin is also an adipokine.
  • I consider that this section should be entitled “White adipose tissue secreted molecules”, because all factors are secreted by white adipose tissue. Similarly, I would encourage authors to include a section regarding “Brown adipose tissue secreted molecules”, because of the increasingly evidences of the importance of the secretory role of brown adipose tissue and the contribution of batokines in regulating systemic metabolism (i.e. FGF21, CXCL14, Mtrnl, BMPs, NRG4…), and how the identification of these molecules can drive researchers to the development of pharmacological treatments against obesity targeting their actions.

Minor aspects:

  • In line 497, the sentence “… the appetite-suppressants described above.” is somewhat confusing because orlistat is the first drug described in this section.
  • I would suggest authors to move table 1 to a closer position to drugs section, maybe before “Other drugs in potential use for anti-obesity treatment”.

Author Response

Response to Review Comments

We are very much thankful to the referee for the deep and thorough review. We have revised the present review paper in the light of the suggestions and comments. We think that this revision has improved the paper and hope that this meets the level of the reviewer’s expectation. Number wise answers to the specific comments/suggestions are as follows:

Comments and Suggestions for Authors

In the current manuscript entitled “Obesity – an update on the basic pathophysiology and review of recent therapeutic advances”, Gjermeni and collaborators review the main pathological mechanisms involving the development of obesity, including a clear summary of the main endocrine factors and their action on the CNS, participating in the regulation of energy balance. The authors also include an analysis of the state-of-the-art on the currently approved pharmacological strategies to treat obesity.

The review is well-structured and provides very complete information, and under my opinion, very few criticisms can be done, although I would like to suggest some recommendations to clarify some aspects of the manuscript, especially involving the section “Adipose tissue secreted molecules”:

  • In the introductory paragraph, I would suggest that authors mention that the adipokines described in that section are only some of the several adipokines currently described (i.e. adipokines not included are omentin-1, SFRP5, cardiotrophin, FABP4, lipocalin-2, apelin…) and the reason of this selection.

We thank the reviewer for his suggestion. Since we have now added a new paragraph including molecules secreted by the brown adipose tissue, we now provide a general description of adipokines and clearly state that we do not exhaustively deal with the topic and we point to a recent review.

  • I intuit the reason why authors highlight the importance of leptin in regulating energy balance and they do not include it in this section, but authors should clearly indicate that leptin is also an adipokine.
    • We highlighted in the beginning of leptin description that it is also an adipokine. We now also mention leptin in “White adipose tissue secreted molecules” section.

  • I consider that this section should be entitled “White adipose tissue secreted molecules”, because all factors are secreted by white adipose tissue. Similarly, I would encourage authors to include a section regarding “Brown adipose tissue secreted molecules”, because of the increasingly evidences of the importance of the secretory role of brown adipose tissue and the contribution of batokines in regulating systemic metabolism (i.e. FGF21, CXCL14, Mtrnl, BMPs, NRG4…), and how the identification of these molecules can drive researchers to the development of pharmacological treatments against obesity targeting their actions.
    • Thank you for pointing out the relevance of brown adipose tissue. Following the reviewers' suggestion, we included a section entitled "Brown adipose tissue secreted molecules" in the Adipose tissue secreted molecules section. We briefly summarized the potential of adipose tissue browning for obesity intervention and described a selection of brown adipose tissue secreted factors. Besides this we also referred to recent reviews focusing on the topic.
    • We also changed the title of the previous section as suggested.

Minor aspects:

  • In line 497, the sentence “… the appetite-suppressants described above.” is somewhat confusing because orlistat is the first drug described in this section.
    • We thank the reviewer for the comment. This sentence was changed to “the appetite-suppressants described in the following sections”
  • I would suggest authors to move table 1 to a closer position to drugs section, maybe before “Other drugs in potential use for anti-obesity treatment”.
    • We moved table 1 before “Other drugs in potential use for anti-obesity treatment”.

Reviewer 2 Report

The review article by Gjermeni et al., describes that obesity represents a major public health problem with a prevalence increasing at an alarming rate worldwide. In fact, continuous intensive efforts to elucidate the complex pathophysiology and improve clinical management have led to a better understanding of biomolecules like gut hormones, antagonists of orexigenic signals, stimulants of fat utilization, and/or inhibitors of fat absorption. In this review, they describe the pathophysiology and pharmacotherapy of obesity including intersection points to the new generation of antidiabetic drugs. They provide insight into the effectiveness of currently approved anti-obesity drugs and other therapeutic avenues that can be explored.

This is a very broad and well-described review of obesity and its possible therapeutic targets. It is well explained and is quite complete. I have the following comments.

  • The section on SGLT2 inhibitors should be expanded, as well as its cardioprotective capacity.
  • The relationship between obesity, insulin resistance, and mitochondrial dysfunction is not described and should be included in this review.
  • A figure is needed to explain the relationship between mitochondrial dysfunction as a therapeutic target for obesity.
  • Antioxidant treatment should be included in this review as well as a table of approximations to the result.

Author Response

Response to Review Comments

We are very much thankful to the referee for the deep and thorough review. We have revised the present review paper in the light of the suggestions and comments. We think that this revision has improved the paper and hope that this meets the level of the reviewer’s expectation. Number wise answers to the specific comments/suggestions are as follows:

Comments and Suggestions for Authors

The review article by Gjermeni et al., describes that obesity represents a major public health problem with a prevalence increasing at an alarming rate worldwide. In fact, continuous intensive efforts to elucidate the complex pathophysiology and improve clinical management have led to a better understanding of biomolecules like gut hormones, antagonists of orexigenic signals, stimulants of fat utilization, and/or inhibitors of fat absorption. In this review, they describe the pathophysiology and pharmacotherapy of obesity including intersection points to the new generation of antidiabetic drugs. They provide insight into the effectiveness of currently approved anti-obesity drugs and other therapeutic avenues that can be explored.

This is a very broad and well-described review of obesity and its possible therapeutic targets. It is well explained and is quite complete. I have the following comments.

  • The section on SGLT2 inhibitors should be expanded, as well as its cardioprotective capacity.
    • We made significant changes in the section on SGLT2 inhibitors including an overview of the major clinical outcome trial that recently changed the recommendations for clinical practice. We also discuss the wide spectrum of proposed mechanisms that lead to the observed cardio- and renoprotective effect.
  • The relationship between obesity, insulin resistance, and mitochondrial dysfunction is not described and should be included in this review.
    • We included a new paragraph titled “Obesity, mitochondrial dysfunction and insulin resistance – an interplay”.

  • A figure is needed to explain the relationship between mitochondrial dysfunction as a therapeutic target for obesity.
    • We have now included Fig. 2 to show the mechhanisms of mitochondrial dysfunction, which we coupled to the section entitled “Obesity, mitochondrial dysfunction and insulin resistance – an interplay”.
  • Antioxidant treatment should be included in this review as well as a table of approximations to the result.
    • We thank the reviewer for the suggestion. We have now included “Antioxidants” in the section “Further potential pharmacotherapeutic targets”. Because we only provide a table with results for established treatments, but also since we now provide 3 figures and one table, we could not include the antioxidants in a further table. We hope that the reviewer can understand the limitation and find our overview appropriate.

Round 2

Reviewer 2 Report

No more comments